# The Role of Metal Ions in Fungal Organic Acid Accumulation

**DOI:** 10.3390/microorganisms9061267

**Published:** 2021-06-10

**Authors:** Levente Karaffa, Erzsébet Fekete, Christian P. Kubicek

**Affiliations:** 1Department of Biochemical Engineering, Faculty of Science and Technology, University of Debrecen, H-4032 Debrecen, Hungary; kicsizsoka@yahoo.com; 2Institute of Chemical, Environmental & Bioscience Engineering, TU Wien, A-1060 Vienna, Austria; peter.kubicek@tuwien.ac.at

**Keywords:** manganese(II) ions, copper(II) ions, iron(II) ions, zinc(II) ions, organic acid fermentations, filamentous fungi

## Abstract

Organic acid accumulation is probably the best-known example of primary metabolic overflow. Both bacteria and fungi are capable of producing various organic acids in large amounts under certain conditions, but in terms of productivity-and consequently, of commercial importance-fungal platforms are unparalleled. For high product yield, chemical composition of the growth medium is crucial in providing the necessary conditions, of which the concentrations of four of the first-row transition metal elements, manganese (Mn^2+^), iron (Fe^2+^), copper (Cu^2+^) and zinc (Zn^2+^) stand out. In this paper we critically review the biological roles of these ions, the possible biochemical and physiological consequences of their influence on the accumulation of the most important mono-, di- and tricarboxylic as well as sugar acids by fungi, and the metal ion-related aspects of submerged organic acid fermentations, including the necessary instrumental analytics. Since producing conditions are associated with a cell physiology that differs strongly to what is observed under “standard” growth conditions, here we consider papers and patents only in which organic acid accumulation levels achieved at least 60% of the theoretical maximum yield, and the actual trace metal ion concentrations were verified.

## 1. Introduction

Commercial production processes for organic acids are excellent examples of fungal biotechnology and unrivalled in terms of productivity. Some of them are also commodity products and represent-after antibiotics and amino acids-the third-largest category of bulk chemicals manufactured by microbial fermentation [1]. Because of their weak acidity they are optimal for food applications such as flavours and preservatives, but they also function as building blocks for polymer production and suitable platform chemicals in green chemistry. Among a list of over 300 chemicals that can be produced from biomass and have the potential to be transformed into new type of useful molecules, the top twelve include eight organic acids [1]. However, to successfully compete with petroleum-based chemical technologies, bio-based organic acid production processes must still improve in economic efficiency.

The organic acids accumulated by fungi can be distinguished into four types: (a) monocarboxylic acids such as lactic acid; (b) dicarboxylic acids, which comprise oxalic, malic, fumaric, itaconic, succinic and trans-epoxysuccinic acid; (c) tricarboxylic acids such as citric acid; and (d) sugar acids such as gluconic acid [2] (Table 1). With the exception of lactic acid which is an end-product of glycolysis, and gluconic acid which is formed by an extracellular enzyme, all the other acids can be considered as products of incomplete aerobic D-glucose catabolism. This incomplete catabolism has been coined by Foster [3] as ”overflow metabolism”, because it is best observed in the presence of high concentrations of D-glucose and simultaneous limitation in other nutrients such as nitrogen, phosphorous and-at least in the case of citric and itaconic acid-certain divalent metal ions.

The investigation on the influence of trace metals on fungal organic acid accumulation had started a century ago and it is therefore necessary to stress a few points: first: at the time of the first attempts to optimize culture media for production, it was not yet known which trace metals would be essential for fungi. For this reason, many studies were performed to test whether the addition of trace metals would enhance rather than decrease organic acid accumulation. Second, until the ground-breaking studies on submerged citric acid fermentation by Shu and Johnson [4,5,6], all studies were performed in surface cultures and in most cases with low concentrations of the carbon source (1–2%, *w*/*v*) [7,8,9,10,11,12,13,14,15,16]. For the citric acid fermentation and its impairment by manganese ions it has later been clearly documented that surface cultivation is much less sensitive to the negative influence of trace elements [17]. Third, most of these early studies (and unfortunately even many of the contemporary ones) were not performed under conditions in which product formation rate and yield could be compared to the industrial organic acid production processes and thus do not allow conclusions to be drawn as to the occurrence of the observed effects under true producing conditions. Finally, many investigations on the effect of trace metals on organic acid fermentations did not consider the fact that the carbon source and the inorganic medium components all introduce traces of metal ions and their results therefore do not clearly demonstrate whether the further addition of a trace metal ion had an effect or not.

For all these reasons we will in this review only consider those papers in which organic acid production was studied under conditions that allowed acid accumulation at least at 60% of the theoretical maximum yield, and in which the actual trace metal ion concentration was verified by chemical analysis.

## 2. Biology of First-Row Transition Metal Ions in Fungi

Four of the first-row transition metals-i.e., manganese (Mn), iron (Fe), copper (Cu) and zinc (Zn)-play major roles in the physiology of fungi. Three of them-Mn, Fe, Cu-provide the necessary redox and catalytic activity for many important biological processes. They possess a stable +2 oxidation state and can generate many additional stable states (up to seven in the case of Mn), which allows them to readily change their oxidation states in biological reactions. Zinc (Zn), with its single oxidation state (+2) and its filled d-orbital, is an exception, but nonetheless plays important roles especially in eukaryotic gene regulation.

Iron is the most abundant of these trace metals in organisms and fulfils a number of essential roles: these include oxygen transport, the tricarboxylic acid cycle, and electron transport chains [18]. In the latter case this occurs via incorporation of iron or the iron-containing prosthetic group heme into the active centres of key enzymes. All these functions are related to its chemical redox properties, i.e., the capacity to readily switch between the ferric (Fe^2+^) and the ferrous (Fe^3+^) forms. This property can, however, also create problems as Fe^3+^, the prevalent form under aerobic conditions, is essentially insoluble in water and is thus inaccessible [19]. Fe^2+^ in contrast is much more soluble but can form radicals via the Fenton reaction and thus cause cellular damage. Finally, iron is required for the enzyme catalase which is essential for the removal of hydrogen peroxide [20] that is a product of the reaction of flavoprotein oxidases.

Zinc is the next most abundant of these four trace metal elements in biological systems and serves as a structural and catalytic co-factor for many proteins, including alkaline phosphatase, Cu-Zn superoxide dismutase, and alcohol dehydrogenases [21]. In addition, it is a component of several regulatory proteins that possess structural domains, such as the zinc finger domain, the LIM domain (named after Lin-11, Isl-1 and Mec-3, where it has been first found [22]), and the RING (“really interesting new gene” [23]) finger domain [24,25]. In fact, about 10% of human genes code for zinc-binding proteins, and about 8% of the yeast proteome is believed to bind zinc [26].

Copper is a different story. Although it is-as with iron and zinc-required as an essential trace element in many biochemical reactions, it rapidly becomes toxic at concentrations where Fe, Zn and Mn do not, since it inactivates other metalloenzymes by metal displacement, and generates reactive oxygen species by the Fenton reaction [27,28,29] This toxicity of copper is also a means for the competition between organisms, e.g., the innate phagocyte utilizes copper for defence against microbes [30,31]. Sándor et al. [32] have recently shown that copper toxicity in *A. terreus* is dependent on the carbon source and the concentration of Mn.

Many Cu-containing enzymes have oxygen-related functions, such as mitochondrial cytochrome c oxidase of the respiratory electron transport chain, and cell wall-bound and cytosolic Cu-dependent superoxide dismutases (SODs) can protect fungal cells from externally and internally generated oxidative stress. Finally, it also has an important function as a cofactor of laccases and tyrosinases [33,34], which are required for the biosynthesis of melanin.

Manganese is required for a small set of enzymes that interact with nucleotides, such as RNA polymerases, adenylate cyclase, pyruvate carboxylase, sugar transferases of the Golgi and also for the mitochondrial Mn-dependent SOD (reviewed for baker’s yeast by Reddi et al. [35]). Manganese ions have also been described to be required as a cofactor for several dehydrogenases and kinases, but this effect is likely physiologically irrelevant: the same stimulating effect can be seen with magnesium ions, because these enzymes require a divalent metal ion-nucleotide complex as substrate, and Mg^2+^ is preferred because it is present in much higher cellular concentrations (1–100 mM; [36]).

## 3. The Influence of Transition Metal Ions on Accumulation of Mono-and Dicarboxylic Organic Acids by Fungi

As explained earlier, the influence of metal ions on organic acid accumulation was investigated in the mid-previous century, when industrial fermentations known at that time were citric-, fumaric-, lactic- and itaconic acid. Because of the very similar biosynthesis of itaconic acid and citric acid [37], we will treat them together in the next chapter.

Interest in the production of the other-particularly dibasic-organic acids rose only much later. This difference is reflected in the fact that-with the exception of fumaric- and itaconic acid-the dicarboxylic organic acids have so far not been thoroughly investigated with respect to a potential influence of metal ions. A notable exception is the study by Battat et al. [38], who reported that an increase of the concentration of iron salts in the medium up to 12 mg L^−1^ enabled *A. flavus* to accumulate 113 g malic acid from 120 g D-glucose. While this sounds like a high yield, one must however bear in mind that the stoichiometry of malic acid production from glucose is
1 Glucose + 2 CO_2_ (supplied as CaCO_3_) → 2 Malic acid.
as in [2], and thus 120 g glucose would allow a theoretical yield of 180 g malic acid. The concentration accumulated by *A. flavus* in the above example is therefore only 62.7% of the theoretical yield, indicating that there is still room for improvement. Most of the research on the dibasic organic acids has been dedicated to increase the yield by genetic manipulation (for review see [2]). An *A. oryzae* strain, in which several genes related to glucose transport and catabolism had been overexpressed, exhibited the highest malic acid yield so far reported (80% of the theory; [39]). In this paper, the authors apparently did not consider metal ion concentrations in the medium, and it is thus possible that L-malic acid production is not affected by divalent transition metal ions.

Nothing is known about whether transition metal ion concentrations would affect the production of succinic and *trans*-epoxy-succinic acid. Published studies do not indicate whether a potential role of trace metal ions has ever been considered, but the reported yields were always very low (around 20% of the theoretical yield), thus precluding any conclusions.

On the other hand, zinc-but not Fe, Mn or Cu-has been shown to be critical for fumaric acid accumulation by *Rhizopus nigricans* [40]. In the absence of added zinc, *Rhizopus* converted a large part of glucose into fumaric acid, accompanied by relatively slow and poor growth, whereas the presence of 1.2 mg L^−1^ of zinc ions in an otherwise complete nutrient medium favoured good and fast growth. The authors also noted that the effect of zinc deficiency became stronger when *R. nigricans* grew on high concentrations of glucose (>10%, *w*/*v*). More recently, Ge et al. [41] confirmed these findings in *R. oryzae* and explained it by the fact that its lactate dehydrogenase requires zinc as a cofactor and a limitation in this metal ion reduces the by production of lactic acid and thus increases the fumaric acid yield. This is in agreement with the early findings of Lockwood [42] that lactic acid production in *R. oryzae* is enhanced by adding up to 10 mg L^−1^ ZnSO_4_. 7H_2_O.

## 4. The Influence of Transition Metal Ions on Accumulation of Citric- and Itaconic Acid by Fungi

The model case, in which the concentration of certain divalent transition metal ions is critical for organic acid accumulation by fungi, is citric acid. Pioneering efforts in this direction were published by Shu and Johnson [4,5,6] using *A. niger* ATCC 11414 in submerged fermentation with 14% (*w*/*v*) glucose as a substrate. To this end, they purified the glucose solution by aluminium hydroxide co-precipitation, and subsequently added 0.3 mg L^−1^ Zn^2+^ and 1.3 mg L^−1^ Fe^2+^ ions and after 9 days of cultivation obtained a specific molar citric acid yield (Y_p/s_) of 63% of the theoretical maximum. Higher concentrations of these two metal ions decreased the yield, but not below a Y_p/s_ of 15%. Mn^2+^ ions inhibited citric acid accumulation at a significantly lower concentration, and decreased the Y_p/s_ to 15% at 6 µg L^−1^. Similar results were obtained with *A. niger* N-548, which was proprietary to Kyowa Fermentation Industry, Tokyo [43], thus showing that the metal ion sensitivity is not just a rare property of one particular strain. These authors were also the first who replaced the purification of glucose with aluminium hydroxide co-precipitation by ion exchange treatment. Consistent results were obtained by Clark et al. [44] who pioneered the use of ferrocyanide-treated beet molasses as a carbon source for citric acid fermentation by yet another strain of *A. niger*, NRC A-1-233 (=ATCC 26550). The addition of as little as 2 ppb (=μg L^−1^) of Mn^2+^ to ferrocyanide-treated beet molasses during citric acid fermentation caused a 10% reduction in Y_p/s_, and they also observed that the addition of Mn^2+^ caused an undesirable change in the morphology from the pellet-like form typical for high-yield citric acid fermentations to the filamentous form. Even smaller doses (0.4–2 ppb) caused undesirable pellet clumping, while greater additions (2–100 ppb) resulted in further decreases in citric acid yield, down to 15% of the theoretical Y_p/s_. The addition of Fe^2+^, Zn^2+^, or Cu^2+^ did not affect the citric acid yield. Relevant literature regarding the effect of Mn^2+^ and Fe^2+^ ions on citric acid molar yield is summarized in Figure 1a,b.

While these studies clearly identified the extracellular concentration of Mn^2+^ ions as a critical determinant of citric acid production, some articles claimed an inhibitory effect of iron: Szczodrak and Ilczuk [50], working with potassium ferrocyanide-treated molasses, found that the addition of 200 mg L^−1^ of Fe^2+^ decreased Y_p/s_ to 10%. One should not forget that iron salts of even analytical grade-as used in research-may contain up to 0.5% (w/w) manganese ions as impurities. This fact implies that the addition of 200 mg L^−1^ iron introduces a concentration of 1 mg L^−1^ of Mn(II) ions into the culture broth, which would obviously inhibit citric acid formation. On the other hand, Odoni et al. [51] working with a low sugar concentration (2%, *w*/*v*) added 0.04 mg L^−1^ of FeSO_4_.7H_2_O and observed an about 60% decrease in citric acid accumulation. It is possible that this effect only occurs at very low sugar concentrations, because similar results were also reported for Zn^2+^ [52,53]. These authors-using a sugar concentration of 0.4–0.8% (*w*/*v*)-showed that citric acid only accumulated when the concentration of zinc was maintained at <1 µM, and the addition of zinc to citric acid-accumulating cultures resulted in the reversion to biomass formation at the expense of acidogenesis. Cyclic AMP (cAMP) affected both the rates of growth and acidogenesis when added to cultures growing at low but not at high zinc(II) ion concentrations. cAMP augmented citric acid synthesis when added to already accumulating mycelia. In the cultivation system applied in these studies iron, manganese, or calcium at concentrations as high as 5–10 µM had no influence on either growth or citrate accumulation. We should note that the sugar concentration in the medium is important for citric acid accumulation not only because more sugar can then be converted to citrate, but because an increase in the concentration of the sugar increases specific molar yield (Y_p/s_; [54]). The effect of Zn^2+^ and cAMP could not be confirmed in the presence of the standard 14% (*w*/*v*) sucrose [5,55].

With regards to itaconic acid accumulation, relatively few studies have been published that dealt with a potential influence of divalent transition metal ions. Ferrocyanide treatment was shown to enhance itaconic acid production [56], which suggests that the metal ions thereby precipitated may decrease the itaconic acid yield. Evidence was provided that the overproduction of itaconic acid requires a medium in which the Mn^2+^ concentration is limiting and in the same range as in citric acid fermentations [57,58]. Increasing the concentration of Mn^2+^ ions up to 100 µg L^−1^ halved the itaconic acid yield. Similar to the findings with citric acid fermentation [6], lowering the concentration of inorganic phosphate alleviated the inhibitory effect of Mn^2+^ on itaconic acid production [58]. Interestingly, however, there are also reports on itaconic acid fermentations with high yields which do not attempt to remove metal ions [59,60]. This issue therefore needs further investigation.

## 5. Biochemical and Physiological Interpretations of the Roles of Manganese Ions on Citric and Itaconic Acid Accumulation

Manganese deficiency has multiple effects on the fungal physiology and can, for example, result in an increased protein turnover [61], impaired DNA biosynthesis [62] and altered composition of plasma membranes [63] and cell walls [64], which indicates that the manganese effect is not clearly related to a single cellular function. Interestingly, all these symptoms are also typical for two major cellular processes: oxidative stress and apoptosis. Under conditions of manganese deficiency, expression of MnSOD is severely inhibited leading to severe oxidative stress indicated by e.g., increased protein turnover and changes in plasma membrane lipid composition [65,66]. Increased protein turnover is also typical for autophagy [67], and a knockout of two autophagy genes in *A. niger* decreased the accumulation of citric acid [68]. Deficiency in zinc(II), manganese(II) or iron(II) has been shown to induce autophagy in *A. fumigatus* [69]. Oxidative stress and apoptosis therefore constitute an interesting field for further studies towards understanding the biological role of manganese(II) ions in *A. niger* and *A. terreus.*

The similar effect of manganese(II) ions on citric and itaconic acid production indicates that the cellular processes involved must be critical for the accumulation of both acids. Consequently, this provides the chance to re-evaluate the different biochemical consequences that have been attributed to manganese deficiency for citric acid accumulation: the eldest one is the hypothesis that the flux rate through glycolysis may be stimulated because manganese ion deficiency enhances the pool of NH_4_^+^ ions, which is capable of antagonizing the inhibition of PFK1 by citrate [70]. Enhanced intracellular concentrations of citrate under manganese ion deficiency have been demonstrated [71], and their localization in the cytosol is likely because citrate has to pass through it before being transported out of the cell. During itaconic acid accumulation, however, it is itaconate that accumulates in the cytosol, and the concentration of citrate should not be significantly higher than under non-producing conditions. It is not known whether itaconate inhibits PFK1, but it has been shown to mimic the inhibition of PFK2 by phosphoenolpyruvate (PEP) [72]. Since PEP is also an inhibitor of PFK1, it is possible that itaconic acid also exerts inhibition on the glycolytic flow at PFK1. In fact, transformation of *A. terreus* with a gene encoding a citrate-resistant fragment of PFK1 stimulates the rate of production and volumetric yield of itaconic acid [73], indicating the importance of the glycolytic flux rate for itaconic acid fermentation, too.

Another explanation for the effect of manganese ion deficiency on the accumulation of citric acid was the necessity to avoid the re-uptake of citrate that would obviously decrease external accumulation. Uptake of citrate in the production phase of the fermentation was indeed shown for *A. niger* [74]. Since manganese-chelated citrate can be imported by *A. niger* cells, by preventing the formation of manganese-citrate complexes, re-uptake can be averted and a higher productivity be obtained [75]. In *Saccharomyces cerevisiae,* Mn^2+^ ions can also be imported in complex with phosphate via the Pho84 transporter [76]. *A. niger* has a corresponding ortholog (NRRL3_00737; CBS 513.88: ANI_1_1172124; ATCC1015: ASPNIDRAFT 121846), and its operation would be indirectly supported by studies showing that the detrimental effect of Mn^2+^ on citric acid accumulation can be decreased by lowering the concentration of inorganic phosphate in the culture broth [5].

While itaconic acid is also capable of forming complexes with divalent metal ions, they lack the OH^−^ and one COO^−^ group that participate in metal binding by citrate, and the dissociation constants for complexes between itaconic acid and manganese are not known [77]. Although the genes for the citrate and itaconate transporters in *A. niger* and *A. pseudoterreus* have been cloned and analysed [78,79,80], the potential effects of manganese ions on these two permeases have not been studied as yet. The role of manganese deficiency in the transport of tricarboxylic acids remains therefore unclear.

Manganese deficiency has a strong effect on *A. niger* morphology. Mycelia grown under manganese-deficient conditions are strongly vacuolated, highly branched, contain strongly thickened cell walls and exhibit a bulbous appearance [64], and these phenomena may be related to a loss of orientation of apical growth. This results in the characteristic pellet morphology which is very critical for citric acid fermentation and which can be used to monitor and control the fermentation by e.g., automatic image analysis [81]. However, this impugns the interpretation of the effects observed under manganese deficiency: since the effect of Mn^2+^ ions cannot be experimentally separated from its effect on morphology, it must be questioned whether all the effects mentioned above are primary responses. Fermentation broth containing Mn^2+^-deficient mycelium has a much better rheology and oxygen transfer than the one containing the hairy, filamentous pellets formed in the presence of sufficient amounts of Mn^2+^. These pellets result in a considerably improved rheology of the culture broth and thus increased mass transfer [82]. In addition, the oxygen concentration in fungal pellets decreases rapidly with depth >0.8 mm, and the centres of larger pellets might be starved of oxygen and autolyse [83]. Since high oxygen tension in the medium is critical for achieving high yields of citric and itaconic acid [84,85,86], it is likely that manganese ion deficiency has a major role in guaranteeing it. Gyamerah [87] showed that *A. terreus* pellets <1 mm in diameter are enhanced producers of itaconic acid. Thus, the primary roles of Mn^2+^ deficiency in citric- and itaconic acid accumulation are still unclear. Dai [88], using suppression subtractive hybridization, identified 22 genes whose expression responded to Mn^2+^. These differentially expressed genes, which have a tentatively identified function, could be assigned to two general categories: those involved in amino acid or protein metabolism (i.e., in cell growth) and those involved in cell regulation. They concluded that the rapid hyphal growth associated with the switch to filamentous morphology observed upon induction by sufficient Mn^2+^ levels probably requires an increased protein production as well as the degradation and utilization of proteins required for the maintenance of the pellet-type morphology. A functional analysis of one of the unknown genes-*brsa-25*-indicated that it was indeed involved in the regulation of fungal morphology.

## 6. Metal Ions-Related Aspects of Submerged Organic Acid Fermentations

As discussed above, Mn^2+^ ion concentrations higher than 5.6 µg L^−1^ decrease the final yield of citric acid in *A. niger* [64]. This small amount can unintentionally be introduced into the culture broth. In fact, with wide local variations, even tap water may contain up to 3 µg L^−1^ Mn^2+^ ions [57]. Furthermore, the surface of carbohydrates that serve as carbon sources (mostly glucose or sucrose) adsorb divalent cations very well and are then moved into solution by the aqueous solvent (the lower trace metal content is the reason why beet molasses is preferred over cane molasses for citric acid production; [89]). A 1% (*w*/*v*) solution of D-glucose already contains up to 10 µg L^−1^ Mn^2+^ ions (i.e., twice as much as the threshold), and 15% (*w*/*v*) D-glucose exceeds this value 30-fold [57]. Furthermore, the bioreactor itself is an often overlooked source of metal ions: on a technical scale, the stirrer attachment, the aeration system and the sampling tubes of fermenters are all built of stainless steel alloy that may contain up to 2% of manganese [90]. The high temperature during sterilization, the acid pH caused by organic acid accumulation and the sheer force inflicted by the mechanical agitation may corrode the steel surfaces and thus lead to the leaching of metal ions into the culture broth. The extent of corrosion-driven leaching could easily surpass the critical levels even in the initial stage of the fermentations (Fekete, Kubicek and Karaffa; manuscript in preparation).

Various methods have therefore been patented or published to remove the surplus iron and manganese ions from the growth medium. At the laboratory scale, this can be counteracted by passivation (the electrochemical application of an oxide-rich chromium layer) of the fermenter, but this is not applicable on the industrial scale. The most widely used technique-cation exchange of the already dissolved carbon source-uses silica gels of 100–200 Mesh grade that are suitable for gravitational chromatography, so the solvent can flow smoothly without external pressure. Metal ions can also be removed by precipitation with the low-toxicity ferrocyanide (hexacyanoferrate; [91]), or chelation by phytic acid [92]. However, these methods require careful optimization as they are not selective for Mn^2+^ or Fe^2+^, and thus may also decrease the concentration of other important cations (Mg^2+^, Ca^2+^) in the liquid.

A more convenient strategy is therefore to counteract the effect of manganese ions. It has long been demonstrated that the addition of *n*-propanol, ethanol and, in particular, methanol stimulates the production of citric acid by *A. niger*, and higher levels of zinc, iron and manganese are tolerated if a slightly toxic concentration of these lower alcohols are present in the culture broth [93]. Alcohols have been shown to act principally on membrane permeability by affecting phospholipid composition, so their antagonizing effect on metal ions has been attributed to an effect on membrane transport. Citric acid yield can also be improved through the manipulation of membrane permeability by supplementing a variety of quaternary ammonium compounds or amine oxides to the fermentation medium in amounts from about 5 to about 50 parts per million (= mg L^−1^) when Fe^2+^ concentration is over 0.2 mg L^−1^ [94]. The biochemical basis for the effect of membrane affecting agents for counteracting the effect of manganese ions on citric acid production is not known yet; however, recent studies demonstrated that they evoke oxidative stress [95,96] and therefore mimick one of the major effects of manganese deficiency. 

Both for *A. niger* citric- and *A. terreus* itaconic acid fermentations, low product yield in the presence of high Mn^2+^ ion concentrations can be counteracted by increasing the Cu^2+^ concentration in the culture broth, possibly by interfering with cellular Mn^2+^ ion uptake and homeostasis [62,75] or-as with the membrane effecting compounds-by inducing oxidative stress. As explained above, an excess of copper(II) induces the accumulation of oxygen radicals by the Fenton reaction which are-in the presence of manganese ions-counteracted by the activity of a manganese-dependent superoxide dismutase (MnSOD). We thus consider it possible that-besides inhibiting manganese transport-Cu^2+^ can induce the same symptoms of oxidative stress in the presence of manganese that manganese deficiency would cause. In line with this, it was recently described that the ratio of Cu^2+^ to Mn^2+^-rather than their respective absolute concentration-determines micro- and macro-morphology of the *A. terreus* biomass and modulates itaconic acid production yield [32] on the glycolytic carbon sources D-glucose and D-fructose. Inclusion of the Cu^2+^ ion concentration amongst the process variables even led to the formulation of growth media for the production of itaconic acid without the need to remove surplus Mn^2+^ in advance [58].

In contrast to these standard chemical engineering methods that have been well-practised in the industry for several decades now, the identification of the major manganese transporter DmtA of *A. niger*, whose knock-out results in Mn^2+^-insensitive production of citric acid [49], unfolded a novel strategy to address the problem. Furthermore, manipulation of *dmtA* challenged the notion that the effect of Mn^2+^ deficiency on citric acid accumulation and hyphal morphology is a consequence of insufficient availability of this metal ion. Cultivation of a *dmtA*-overexpressing strain at 5 µg L^−1^ Mn^2+^ ions produced the phenotypes of manganese sufficiency (low citric acid yield, high growth rate, filamentous morphology). This finding suggests that intracellular Mn^2+^ sufficiency mediated by the increased intake rate is more important than the concentration of Mn^2+^ in the medium in prompting the “manganese effects” in *A. niger*.

## 7. Metal Ion Analysis in Organic Acid Fermentations

Determination of metal ion concentrations from the complex matrix formed by the culture broth in mg L^−1^ (iron, copper) or even μg L^−1^ (manganese) ranges requires specialized analytical tools.

Ion chromatography-a form of liquid chromatography-measures concentrations of ions in the mg L^−1^ range. Separation is based on the differential absorption of ions on a resin with anionic functional groups, leading to different retention times. Typically, optical or electrochemical detectors are used to quantify the various ions. The concentration of an ion is represented by the height or the area of the corresponding chromatographic peak. Since specific interactions are involved during the separation, the matrix tolerance is higher.

Ions at the µg L^−1^ range can be determined by inductively coupled plasma-mass spectrometry (ICP-MS). In this case, an inductively coupled plasma-created in an argon stream-efficiently ionizes the sample at 5000–10,000 °C and a mass spectrometer is used as detector [97]. The ions created from the isotopes of the same element are directed into the MS unit which sorts the ions according to their mass/charge ratio. In the detector, the ions generate electrons by colliding with a dynode in an electron multiplier tube which quantifies each ion. Although this method is selective and sensitive (down to ng L^−1^), the matrix components often interfere with the determinations, requiring substantial sample pre-treatments.

## 8. Conclusions

Organic acid overflow requires a unique combination of several unusual nutrient conditions such as excessive levels of carbon source, H^+^, and dissolved oxygen, and suboptimal concentrations of phosphate as well as certain trace elements which synergistically influence the yield. Of this latest group, the importance of four first-row transition metals-copper(II), iron(II), zinc(II) and, most notably, manganese(II)-stand out. Their general role in (fungal) physiology is to provide redox and catalytic activity for a variety of important biochemical reactions as enzyme co-factors. Some of these enzymes-directly or indirectly-play key roles in the biosynthesis of organic acids; thus, manipulation of co-factor supply via careful adjustment of the concentration of these metals can fundamentally change flux distributions in the cell, prompting massive metabolic overflow events. Since organic acid production yields-with the exceptions of citric- and itaconic acid-are still far from the theoretical maxima, the quest for further improvement must inevitably include these trace metals and their roles in metabolism.

## Figures and Tables

**Figure 1 microorganisms-09-01267-f001:**
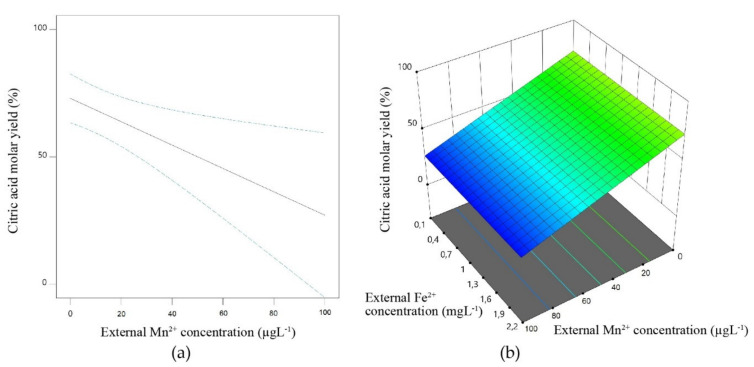
Response surface-based demonstration of the relationship between citric acid molar yield and the external concentration of metal ions. Input data are taken from well-documented, publicly available fermentations [4,6,45,46,47,48,49]. Plots were made with Design Expert 11 (Stat-Ease Inc., Minneapolis, MN, USA). (**a**) 2D plot showing the effect of external Mn^2+^ ion concentration on citric acid molar yield. Dashed lines indicate uncertainty margins. (**b**) 3D plot showing the interaction effects of external Mn^2+^ and Fe^2+^ ions on citric acid molar yield.

**Table 1 microorganisms-09-01267-t001:** Microbial producers and applications of organic acids produced partially or completely by biological means *.

Organic Acid	Producer Micro-Organism(s)	Applications
L-lactic acid	*Lactobacillus* spp. *Lactococcus* spp. *Rhizopus oryzae*	Excipient in food, cosmetics, pharmaceutical and chemical industries. Building block for poly-lactic acid.
Oxalic acid	*Aspergillus niger*	Food industry, pharmaceuticals, waste water treatment, hydrometallurgy
L-malic acid	*Aspergillus niger* *Aspergillus oryzae* *Rhizopus oryzae*	Nutritional bars, protein drinks, functional beverages, pharmaceutical, cosmetic and personal care products
Fumaric acid	*Rhizopus oryzae*	Food acidulant, mordant for dyes. Its esters are used to treat relapsing-remitting multiple sclerosis and have immunomodulating activities.
Succinic Acid	*Saccharomyces cerevisiae* *Escherichia coli* *Anaerobiospirillum succiniciproducens* *Actinobacillus succinogenes* *Basfia succiniciproducens*	Food additives, detergents, pigments, toners, cosmetics, cement additives, pharmaceuticals, resins coatings Building block for 1,4-butanediol, poly-butyl succinate, polybutylene succinate-co-butylene terephthalate.
Trans-2,3-Epoxysuccinic acid	*Aspergillus clavatus* *Paecilomyces varioti* *Paecilomyces elegans* *Penicillium vineferum* *Talaromyces wortamannii*	Building block for optically specific single β-lactam antibiotics and polyepoxysuccinic acid plasticizer stabilizer, corrosion inhibitor, biotransformation to meso-tartaric acid.
Citric acid	*Aspergillus niger*	Flavoring agent and preservative in food and beverages. Emulsifying agent. Delivery of metal ions as citrate salts in dietary supplements. Acidifier and chelating agent in the chemical and pharmaceutical industries.
Itaconic acid	*Aspergillus terreus*	Pharmaceutical, architectural, paper, paint, and medical industries as plastics, resins, paints, synthetic fibers, plasticizers, detergents Drug delivery in medicine.
Gluconic acid	*Gluconobacter* sp. *Aspergillus niger*	Flavoring agent in meat, wine and dairy products. Counterion during therapeutic calcium and/or iron administration. Removal of calcareous and rust deposits from surfaces.

* only where commercially relevant fungal production exists. For this reason, acetic acid, propionic acid and butyric acid, exclusively produced by bacteria (*Acetobacter* spp., *Propionibacter* spp. and *Clostridium* spp., respectively) are omitted.

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
