# Peer review of "The Role of Metal Ions in Fungal Organic Acid Accumulation"

_microorganisms, 2021, doi:10.3390/microorganisms9061267_

Round 1

Reviewer 1 Report

Evaluation and comments to the manuscript No. microorganisms-1255207 entitled “The role of metal ions in fungal organic acid accumulation”.

Authors: Levente Karaffa et al.

In my opinion, this is a very interesting manuscript that is worth publishing in the Microorganisms journal. However, not in its current form. The manuscript must be improved before being accepted for printing.

My comments

1) Please enrich / correct citations in the manuscript. The work contains many fragments of text without references to the literature.

2) Table 1:

 - why is “sp” italicized?

- why are you replacing the bacteria? The work is about fungi.

3) Please improve the quality Fig. 1.

4) In my opinion, the manuscript should be supplemented with a summary.

Author Response

First, we thank both Reviewers for their efforts to improve the presentation of our work.

Reviewer I.

In my opinion, this is a very interesting manuscript that is worth publishing in the Microorganisms journal. However, not in its current form. The manuscript must be improved before being accepted for printing.

My comments

1) Please enrich / correct citations in the manuscript. The work contains many fragments of text without references to the literature.

Reply: We have done so by adding 16 new references.

2) Table 1:

 - why is “sp” italicized?

Reply: We have corrected this.

- why are you replacing the bacteria? The work is about fungi.

Reply: Some organic acids reviewed in this paper are produced both by fungi and bacteria, and we consider this information noteworthy despite of the fungal focus of the review. See also our reply to Comment 3 from Reviewer II.

3) Please improve the quality Fig. 1.

Reply: We have increased resolution. The figures are now uploaded separately, too, to enable technical staff re-enbedding them into the Word file, if needed.

4) In my opinion, the manuscript should be supplemented with a summary.

Reply: We have prepared a Conclusions section.

Reviewer 2 Report

This paper presents a critical and detailed study of the influence of four of the first-row transition metal elements, manganese (Mn2 +), iron (Fe2 +), copper (Cu2 +) and zinc (Zn2 +) on the production of organic acids by fungal species. According to the References section, the first researches carried out in this field were studied, starting from 1947.

The first part of the manuscript shows the producing microorganisms, the uses of the main synthesized organic acids and mentions the aim of this study.

The metabolic role of each metal element and the influence of these metal ions on the accumulation of mono- and dicarboxylic organic acids, as well as citric and itaconic acid by fungi are further described. The effect of metal ions on submerged organic acid fermentations was analyzed, especially in the cultures of Aspergillus niger and A. terreus.

Finally, some aspects regarding metal ion analysis in fungal cultures that produce organic acids are described.

Some observations

In the Abstract - it is debatable whether "in terms of productivity - and consequently, of commercial importance - fungal platforms are unparalleled" it might be appropriate to reformulate this statement

In Table 1 “Lactobacillus sps. , Lactococcus sps. Rhizopus sp. “- the abbreviation sp should not be written in italics

In Table 1 - it would also be good to mention acetic acid as an organic acid produced by the genus Acetobacter

Line 88 - “the ferric (Fe2 +) and the ferrous form (Fe3 +)“  - correct: Fe II is ferrous, Fe III is ferric

Line 156 - I do not think it is necessary to specify (= mg L-1) if ppm is specified

164 - "ZnSO4.7H2O" - the corresponding symbol should be used before 7

176 - it should be checked if it is correct "15% already at 0.6 μg L-1" - or mg L-1?

221 - “specific molar yield (Yp / s; [40]” - a parenthesis is missing

403 - “with a dynode” - correct - dyode

In the end, it would have been good to present some short conclusions

Author Response

First, we thank both Reviewers for their efforts to improve the presentation of our work.

Reviewer II.

This paper presents a critical and detailed study of the influence of four of the first-row transition metal elements, manganese (Mn2 +), iron (Fe2 +), copper (Cu2 +) and zinc (Zn2 +) on the production of organic acids by fungal species. According to the References section, the first researches carried out in this field were studied, starting from 1947.

The first part of the manuscript shows the producing microorganisms, the uses of the main synthesized organic acids and mentions the aim of this study.

The metabolic role of each metal element and the influence of these metal ions on the accumulation of mono- and dicarboxylic organic acids, as well as citric and itaconic acid by fungi are further described. The effect of metal ions on submerged organic acid fermentations was analyzed, especially in the cultures of Aspergillus niger and A. terreus.

Finally, some aspects regarding metal ion analysis in fungal cultures that produce organic acids are described.

Some observations

1) In the Abstract – it is debatable whether "in terms of productivity - and consequently, of commercial importance – fungal platforms are unparalleled" it might be appropriate to reformulate this statement.

Reply: We stand by this statement. No other platforms are capable of converting the carbon substrate into organic acid at molar yields >90% at technical scale. While some organic acids are indeed produced solely by bacteria – see our reply to Comment 3 – this sector of industrial biotechnology overwhelmingly uses fungal platforms.

2) In Table 1 “Lactobacillus sps. , Lactococcus sps. Rhizopus sp. “- the abbreviation sp should not be written in italics.

Reply: We have corrected this.

3) In Table 1 - it would also be good to mention acetic acid as an organic acid produced by the genus Acetobacter

Reply: We have inserted a footnote to this effect.

4) Line 88 - “the ferric (Fe2 +) and the ferrous form (Fe3 +)“  - correct: Fe II is ferrous, Fe III is ferric

Reply: Thank you for noting this; it is now corrected.

5) Line 156 - I do not think it is necessary to specify (= mg L-1) if ppm is specified.

Reply: Non-SI ppm is now deleted.

6) Line 164 - "ZnSO4.7H2O" - the corresponding symbol should be used before 7

Reply: With all due respect, we do not understand this comment. We believe the compound name is correctly written.

7) Line 176 - it should be checked if it is correct "15% already at 0.6 μg L-1" - or mg L-1?

Reply: Thank you for noting this mistake. The value in the original paper is 0.006 mg L-1, that is, 6 mg L-1. It is now corrected.

8) Line 221 - “specific molar yield (Yp/s; [40]” - a parenthesis is missing

Reply: Corrected.

9) Line 403 - “with a dynode” - correct – dyode

Reply: We did mean dynode, a secondary electron multiplier. A key element of all mass spectrometry systems is the detector used to convert a current of mass-separated ions into measurable signal. The essence of the electron multiplier detector is a serial connection of discrete metal plates called dynodes that amplifies a current of ions into a current of electrons. In contrast, a dyode is an electronic component with asymmetric conductance, and has nothing to do with detecting electrons. See reference [97], added newly to the text.

10) In the end, it would have been good to present some short conclusions.

Reply: We have prepared a Conclusions section.

Round 2

Reviewer 1 Report

The authors responded to most of my comments. The current version of the manuscript looks much better than the original version. Good job! Congratulations.